# Salivary Cortisol and Periodontitis Severity: A Cross-Sectional Biomarker-Based Assessment of Stress and Inflammation

**DOI:** 10.3390/medsci13030120

**Published:** 2025-08-08

**Authors:** Velitchka Dosseva-Panova, Antoaneta Mlachkova, Marina Miteva, Dimitar Dimitrov

**Affiliations:** 1Department of Periodontology, Faculty of Dental Medicine, Medical University—Sofia, 1431 Sofia, Bulgaria; v.doseva@fdm.mu-sofia.bg (V.D.-P.); a.mlachkova@fdm.mu-sofia.bg (A.M.); 2Department of Medical Chemistry and Biochemistry, Medical Faculty, Medical University—Sofia, 1431 Sofia, Bulgaria; m.miteva@medfac.mu-sofia.bg

**Keywords:** periodontitis, salivary cortisol, psychological stress, inflammatory biomarkers, interleukin-1β, interleukin-6, risk factors, saliva diagnostics

## Abstract

**Background:** Psychological stress is increasingly recognized as a potential modifier of periodontal disease through both behavioral and biological mechanisms. Cortisol, a key stress hormone, exerts complex immunomodulatory effects and may influence periodontal inflammation and tissue breakdown. This study aimed to compare salivary levels of cortisol, interleukin-1β, and interleukin-6 in patients with varying periodontitis severity and examine their associations with clinical periodontal parameters. **Methods:** A total of 67 patients diagnosed with periodontitis were classified according to the 2017 World Workshop Classification into Stage I/II vs. Stage III/IV and Grade B vs. Grade C. Unstimulated saliva samples were collected and analyzed for cortisol using electrochemiluminescence immunoassay, and for IL-1β and IL-6 using ELISA. Statistical analyses included Mann-Whitney U test, Spearman’s correlations, and multivariate regression. **Results:** Median salivary cortisol levels were significantly higher in Stage III/IV (11.90 nmol/L) than in Stage I/II (7.64 nmol/L; *p* = 0.014) and in Grade C (10.60 nmol/L) vs. Grade B (7.70 nmol/L; *p* = 0.019). In multivariate analysis, cortisol independently predicted both Stage III/IV (OR = 1.23, *p* = 0.007) and Grade C (OR = 1.24, *p* = 0.026) periodontitis. ROC analysis showed that salivary cortisol had moderate diagnostic performance for Stage III/IV periodontitis (AUC = 0.68, cut-off 11.57 nmol/L) and Grade C (AUC = 0.67, cut-off 9.76 nmol/L). Cortisol showed significant positive correlations with clinical markers of disease severity and with IL-1β (r = 0.399, *p* = 0.001) and IL-6 (r = 0.424, *p* < 0.001). **Conclusions:** Salivary cortisol is a promising non-invasive biomarker reflecting both stress-related physiological burden and clinical severity in periodontitis. Cortisol measurement may represent a valuable addition to multifactorial assessments and risk stratification in periodontitis, pending further validation in longitudinal studies.

## 1. Introduction

Periodontitis is a significant global public health concern, affecting a large proportion of the adult population. Its progression can lead to tooth loss, adversely impacting overall health and quality of life and increasing healthcare costs. A study by Chen et al. reported 1.1 billion cases of severe periodontitis worldwide in 2019, underscoring the growing burden of the disease and the urgent need for upstream policy interventions to address this challenge [1]. Although dysbiotic dental biofilm is the primary etiological agent of periodontitis, the severity and progression of the disease are largely driven by the host’s immune inflammatory response, which is influenced by systemic, environmental, genetic, and behavioral factors [2].

The recognition of psychological stress as an emerging risk modifier for periodontal disease is gaining increasing importance in both clinical practice and public health. Stress is believed to affect periodontal health through a combination of behavioral and biological mechanisms. Psychological stress can impair individuals’ attention to oral hygiene, resulting in increased plaque accumulation and the progression of periodontal disease [3]. Stress-related neglect of regular brushing and flossing further exacerbates periodontal conditions [4]. Stress may also contribute to unhealthy lifestyle behaviors, such as increased smoking and poor dietary habits, both of which are established risk factors for periodontal disease [5].

Stress induces the release of cortisol, a glucocorticoid hormone that modulates immune function. Elevated cortisol levels can suppress the immune response, impairing the body’s ability to combat periodontal pathogens and increasing vulnerability to tissue destruction [6]. Cortisol exerts complex immunomodulatory effects. In acute stress, cortisol serves a protective role by limiting excessive inflammation; however, chronic elevation can suppress protective immune response, impair wound healing, and enhance susceptibility to infections [5,7,8,9]. Specifically, cortisol inhibits pro-inflammatory cytokine production, impairs neutrophil chemotaxis and phagocytosis, and alters T-cell function, all of which may weaken the host’s defense against periodontal pathogens and contribute to tissue breakdown [10,11,12,13].

Salivary cortisol has been widely adopted as a non-invasive biomarker of systemic cortisol activity and psychological stress [14,15]. It reflects the biologically active fraction of free cortisol, in contrast to total serum cortisol, and offers practical advantages, including ease of collection, minimal patient discomfort, and reduced risk of stress-induced confounding during sampling [16]. Standardized sampling protocols further allow the control of cortisol’s diurnal variation, enhancing the reliability of salivary assessments in clinical research [17].

Despite the theoretical plausibility linking psychological stress and periodontal disease, empirical findings have been inconsistent. Several studies have reported higher salivary cortisol levels in periodontitis patients compared to healthy controls, suggesting an association between stress and disease severity [8,11,18,19,20,21,22]. However, others have found no significant differences [9,10,23,24], highlighting the methodological challenges inherent in this area of research. Variations in study design, sample characteristics, cortisol measurement techniques, timing of saliva collection, and definitions of periodontal disease contribute to the heterogeneity of findings.

Notably, many earlier investigations relied on outdated or heterogeneous criteria for diagnosing and classifying periodontitis, limiting comparability across studies. The 2017 World Workshop on the Classification of Periodontal and Peri-Implant Diseases and Conditions introduced a more comprehensive system, staging periodontitis based on disease severity and complexity and grading based on biological progression and risk factors [25]. This contemporary framework allows for more standardized categorization and enhances the validity of clinical research on periodontal disease. The proposed definitions aim to incorporate biological features and risk factors, moving towards a precision medicine approach in managing periodontitis [26].

Although the individual relationships between psychological stress, inflammatory cytokines, and periodontitis have been investigated [9,19,27], an integrated analysis that simultaneously evaluates salivary cortisol, interleukin-1β, and interleukin-6 in the context of standardized disease staging and grading remains limited. To the best of our knowledge, only one study has assessed salivary cortisol across periodontitis stages using the 2017 Classification [11], but it did not include inflammatory biomarkers. Considering the complex interplay between stress and immune response in periodontal pathogenesis, a comprehensive assessment of neuroendocrine and inflammatory markers may offer deeper insights into mechanisms driving disease severity. In this study, we focused on IL-1β and IL-6, which are central inflammatory mediators in periodontitis, to provide a representative assessment of the inflammatory response alongside salivary cortisol.

We hypothesize that patients with more severe forms of periodontitis would exhibit higher salivary cortisol, as well as elevated levels of IL-1β and IL-6, reflecting a coordinated neuroendocrine inflammatory response associated with disease progression. Accordingly, the primary objective of our study was to compare salivary cortisol concentrations between different stages and grades of periodontitis. Secondary objectives were to evaluate the correlations between cortisol and the inflammatory cytokines IL-1β and IL-6, and to examine the associations between all salivary biomarkers and clinical periodontal parameters.

## 2. Materials and Methods

This cross-sectional observational study was carried out at the Medical University of Sofia. Ethical approval was obtained from the university’s Research Ethics Committee (protocol No. 06/01.06.2023), and all procedures adhered to the Declaration of Helsinki. All participants provided written informed consent prior to enrollment.

A total of 67 adult patients diagnosed with periodontitis were recruited. The study sample included 35 females and 32 males, aged 26 to 72 years (mean age 51.4 ± 9.4).

Inclusion criteria followed the 2017 World Workshop on the Classification of Periodontal and Peri-Implant Diseases and Conditions [25]: interdental clinical attachment loss (CAL) ≥2 mm detectable at ≥2 non-adjacent teeth, or buccal/oral CAL ≥3 mm with probing pocket depth (PPD) ≥3 mm detectable at ≥2 teeth, with no attribution to non-periodontitis causes.

Participants were categorized into two disease severity groups as follows:The Stage I/II group comprised individuals with CAL ≤4 mm, radiographic bone loss limited to the coronal third (<33%) of the root, and PPD ≤5 mm;The Stage III/IV group included patients with CAL ≥5 mm, bone loss extending to the middle/apical third of the root, PPD ≥6 mm, vertical bone loss ≥3 mm, Class II/III furcation involvement, and tooth loss due to periodontitis or masticatory dysfunction.

Exclusion criteria were pregnancy or lactation; diabetes or autoimmune diseases affecting periodontitis pathogenesis; use of hormonal contraceptives or hormone replacement therapy within the past 6 months; use of systemic antibiotics, corticosteroids, or immunomodulatory agents within the previous 6 months; and periodontal treatment within the past 12 months.

All clinical examinations were performed by two calibrated periodontists. Calibration was conducted on 10 separate patients prior to the study, achieving ≥90% intra- and inter-examiner agreement within ±1 mm for PPD and CAL, with an intraclass correlation coefficient (ICC) of 0.87.

Periodontal parameters were measured with a UNC-15 periodontal probe (Hu-Friedy^®^, Chicago, IL, USA), including Full Mouth Plaque Score (FMPS); Full Mouth Bleeding Score (FMBS); probing pocket depth (PPD)—the distance from gingival margin to pocket base; clinical attachment level (CAL)—the distance from cementoenamel junction to pocket base; bleeding on probing (BoP); furcation involvement using the Hamp classification; and tooth mobility using the Miller Index. Radiographic assessment was performed via periapical radiographs, and the BL/Age ratio was calculated based on the site of greatest bone loss.

Unstimulated whole saliva samples were collected between 9:00 and 11:00 a.m. to control for cortisol’s circadian variation. Patients refrained from eating, drinking, smoking, and oral hygiene for at least one hour prior to collection. After rinsing with water and a 10-min rest, saliva was collected via the passive drool method, yielding approximately 2 mL per participant. Samples were immediately centrifuged at 3000 rpm for 15 min. The supernatant was aliquoted and stored at −80 °C until further analysis.

Salivary cortisol levels were measured using electrochemiluminescence immunoassay (ECLIA) on the Cobas e411 analyzer (Roche Diagnostics^®^, Mannheim, Germany), utilizing ruthenium-labeled cortisol derivatives and biotinylated anti-cortisol antibodies. Assay sensitivity was 0.5 nmol/L, with intra- and inter-assay variability <6%, and analyses were performed in duplicate.

The quantification of interleukin-1β and interleukin-6 was performed using sandwich ELISA kits (BioVendor^®^, Brno, Czech Republic; Cat. Nos. RD194559200R and RD194015200R, respectively) following the manufacturer’s instructions. All samples were run in duplicate, and assays showed coefficients of variation <10%.

Statistical analysis was performed using IBM SPSS version 21.0 (IBM Corp., Armonk, NY, USA). The Shapiro-Wilk test was used to assess normality. Due to non-normal distribution, non-parametric tests were applied. Group comparisons were conducted using the Mann-Whitney U test, and correlations were evaluated using Spearman’s rank correlation coefficient. Multivariate logistic regression was used to identify independent predictors of Stage III/IV and Grade C periodontitis, with all relevant variables included simultaneously in the models. Statistical significance was set at *p* < 0.05.

Inter-group receiver operating characteristic (ROC) curve analyses were performed within the periodontitis cohort to assess the accuracy of salivary cortisol in distinguishing between Stage III/IV and Stage I/II, as well as between Grade C and Grade B periodontitis. The area under the ROC curve (AUC) was calculated for each comparison, and the optimal cut-off values for cortisol were determined using the Youden index, maximizing the sum of sensitivity and specificity.

A post hoc power analysis was performed for the primary outcome (salivary cortisol levels), comparing Stage I/II vs. Stage III/IV and Grade B vs. Grade C groups. The calculated effect sizes (Cohen’s d) were 0.83 and 0.61, respectively, with statistical power of 0.91 for the stage comparison and 0.70 for the grade comparison, indicating that the sample size was sufficient for detecting significant group differences. This analysis was performed using G*Power software (version 3.1.9.7) and further validated in R (version 4.5.0).

## 3. Results

### 3.1. Demographic Characteristics

The study cohort consisted of 67 participants: 32 with Stage I/II and 35 with Stage III/IV periodontitis, as well as 34 classified as Grade B and 33 as Grade C. Table 1 summarizes the demographic characteristics by stage and grade. The mean age was similar across groups, with no statistically significant differences between Stage I/II and Stage III/IV (50.4 ± 10.5 vs. 52.3 ± 8.4 years, *p* = 0.409) or between Grade B and Grade C (52.7 ± 10.1 vs. 50.1 ± 8.6 years, *p* = 0.327). The distribution of females and males did not differ significantly between any of the groups (*p* > 0.05). However, there was a highly significant difference in smoking status between Grade B and Grade C patients, with a markedly higher proportion of smokers in Grade C (78.8%) compared to Grade B (26.5%) (*p* < 0.001). No significant difference in smoking status was observed between Stage I/II and Stage III/IV (*p* = 0.551).

### 3.2. Comparison of Salivary Biomarkers Between Stages and Grades

Significantly increased salivary cortisol levels were observed in individuals with greater periodontitis severity. The median cortisol level in the Stage III/IV group was 11.90 nmol/L (IQR: 7.20–17.30), compared to 7.64 nmol/L (IQR: 6.71–9.70) in the Stage I/II group (*p* = 0.014) (Figure 1A). Similarly, individuals in the Grade C group exhibited higher cortisol levels (median 10.60 nmol/L, IQR: 7.34–15.80) than those in Grade B (median 7.70 nmol/L, IQR: 6.48–9.57; *p* = 0.019) (Figure 1B). These findings indicate that salivary cortisol rises with both clinical severity and the risk of progression in periodontitis.

Comparable trends were observed for inflammatory cytokines. Median salivary IL-1β levels were 257.81 pg/mL (IQR: 183.21–286.46) in Stage III/IV, significantly higher than 220.11 pg/mL (IQR: 170.51–260.59) in Stage I/II (*p* = 0.034). For IL-6, the median concentration was 170.56 pg/mL (IQR: 128.61–197.51) in Stage III/IV versus 158.36 pg/mL (IQR: 117.37–184.09) in Stage I/II (*p* = 0.235). Stratification by grade yielded consistent results: IL-1β and IL-6 levels were both elevated in Grade C compared to Grade B without statistical significance (*p* = 0.064 and *p* = 0.251, respectively).

### 3.3. Multivariate Logistic Regression Predicting Severe (Stage III/IV) and Grade C Periodontitis

In multivariate logistic regression analyses, we assessed the independent associations of demographic, behavioral, and biological factors with two key clinical outcomes: severe periodontitis (Stage III/IV) and Grade C periodontitis.

For severe periodontitis (Stage III/IV), salivary cortisol was the only significant independent predictor (OR = 1.23, 95% CI: 1.07–1.45, *p* = 0.007) (Table 2). This finding indicates that elevated salivary cortisol levels correspond to increased odds of advanced periodontal destruction, independent of age, sex, smoking, and inflammatory biomarkers. No other predictors, including IL-1β, IL-6, age, sex, or smoking status, reached statistical significance in the adjusted model.

For Grade C periodontitis, smoking was identified as the strongest independent risk factor (OR = 13.6, 95% CI: 4.7–61.7, *p* < 0.001) (Table 3). This is consistent with the established role of smoking as a major risk modifier for periodontitis grade in the current classification system and underscores its substantial effect on the risk of a higher rate of progression. Cortisol also remained a significant independent predictor for Grade C periodontitis (OR = 1.24, 95% CI: 1.06–1.56, *p* = 0.026), emphasizing a potential link between stress-related physiological response and increased rate of disease progression. Neither IL-1β nor IL-6 demonstrated significant independent associations with Grade C periodontitis in the multivariate analysis.

Overall, these findings highlight the dominant influence of risk modifiers such as smoking on periodontitis grading and reveal an association between elevated cortisol levels and both severe periodontitis and a higher rate of progression. The lack of independent associations for IL-1β and IL-6 after adjustment suggests that the impact of systemic inflammation may overlap with, or be mediated by, other factors such as stress and smoking.

### 3.4. Receiver Operating Characteristic (ROC) Curve Analysis for Salivary Cortisol

To further assess the potential clinical utility of salivary cortisol, ROC curve analyses were performed within the periodontitis cohort. For distinguishing Stage III/IV from Stage I/II periodontitis, the area under the ROC curve (AUC) was 0.68, with an optimal cut-off value of 11.57 nmol/L (sensitivity 54%, specificity 88%). For distinguishing Grade C from Grade B periodontitis, the AUC was 0.67, and the optimal threshold was 9.76 nmol/L (sensitivity 61%, specificity 76%). These results indicate that salivary cortisol demonstrates moderate discriminative ability for both disease severity and grade (Figure 2).

### 3.5. Correlations Between Salivary Biomarkers and Periodontal Measures

Spearman’s correlation analysis was performed to explore the associations between each salivary biomarker and the various clinical periodontal parameters. To enhance clarity and focus, Table 4 presents only those correlations that reached statistical significance (*p* < 0.05). Specifically, higher cortisol levels were associated with increased plaque accumulation (FMPS), greater proportions of sites with deep periodontal pockets (PPD > 5 mm), more extensive clinical attachment loss (CAL ≥ 5 mm), greater bleeding on probing (BoP), and higher bone loss to age ratio (BL/Age), indicating a consistent relationship between cortisol and indicators of periodontal disease severity (Figure 3). Conversely, cortisol levels were inversely correlated with the percentage of shallow pockets (PPD ≤ 3 mm) and lower attachment loss (CAL 1–2 mm), suggesting that higher cortisol concentrations are linked to more advanced rather than mild periodontal conditions.

A similar pattern was observed for IL-1β, which demonstrated positive correlations with FMPS, FMBS, the percentage of sites with PPD > 5 mm, CAL ≥ 5 mm, BoP, and BL/Age, further supporting its association with both inflammatory burden and tissue destruction. IL-1β was also negatively correlated with PPD ≤ 3 mm and CAL 1–2 mm.

For IL-6, significant positive correlations were found with BoP and BL/Age, while a negative association was observed with PPD ≤ 3 mm. Taken together, these results indicate that all three salivary biomarkers—cortisol, IL-1β, and IL-6—are closely related to indicators of inflammation and periodontal tissue breakdown, with higher biomarker levels reflecting increased disease severity.

Salivary cortisol also showed strong positive correlations with IL-1β (r = 0.399, *p* = 0.001) and IL-6 (r = 0.424, *p* < 0.001), indicating a coordinated elevation of stress and inflammatory biomarkers in patients with more severe disease (Figure 4).

## 4. Discussion

This study investigated the association between salivary cortisol levels and clinical and inflammatory features of periodontitis, providing compelling evidence that cortisol—a central hormone in the stress response—reflects not only psychological burden but also correlates with periodontal disease severity and inflammatory activity.

The integration of clinical periodontal classification, salivary biomarker profiling, and immunoinflammatory mediators in our research allowed us to address critical gaps in the literature. Our multivariate regression analyses directly support the theoretical framework of psychoneuroimmunology by showing that cortisol does not merely correlate but independently predicts the severity of periodontal disease. This significantly contributes to understanding how systemic stress can modulate oral health, even when other known risk factors are adjusted for.

Cortisol, as a biomarker of hypothalamic–pituitary–adrenal (HPA) axis activation, has been consistently linked to stress and systemic disease exacerbation [7,28]. In our analysis, patients with Stage III/IV periodontitis had significantly higher median salivary cortisol concentrations compared to Stage I/II (11.90 vs. 7.64 nmol/L, *p* = 0.014), and similarly, Grade C (rapid progression) cases showed elevated cortisol relative to Grade B (10.60 vs. 7.70 nmol/L, *p* = 0.019). These results are consistent with previous findings by Develioglu et al., who observed higher cortisol levels in severe periodontitis compared to mild or moderate cases [29], and by Lee et al., who reported stepwise cortisol increases from healthy individuals through gingivitis to periodontitis, identifying the disease as an independent predictor of cortisol elevation [8]. The systemic review by Corridore et al. (2023) confirmed the link between chronic psychological stress and periodontal disease severity, citing both behavioral and immunological mechanisms [30].

Our study went beyond simple correlations by employing multivariate logistic regression analysis, which revealed that salivary cortisol was the only significant independent predictor for severe periodontitis (Stage III/IV) (OR = 1.23, 95% CI: 1.07–1.45, *p* = 0.007). This finding underscores the unique role of cortisol as a marker for advanced disease, as its association remained strong even after adjusting for factors such as age, sex, smoking, IL-1β, and IL-6. This significantly enhances the understanding of how cortisol reflects not just the presence of disease, but also its clinical intensity and stage.

In the context of periodontal grade assessment, our multivariate regression analyses confirmed the established role of smoking as a dominant independent risk factor for Grade C (OR = 13.6, *p* < 0.001). Beyond this, we found that salivary cortisol was also a significant independent predictor for Grade C periodontitis (OR = 1.24, *p* = 0.026), highlighting a potential connection between chronic stress and accelerated disease progression. This result is particularly important, as it indicates that cortisol contributes to predicting progression independently of the known effect of smoking.

The inclusion of inflammatory cytokine profiles further strengthens the biological plausibility of our findings. In line with the concept of the stress-inflammation feedback loop, our data revealed strong positive correlations between salivary cortisol and the pro-inflammatory cytokines IL-1β (r = 0.399, *p* = 0.001) and IL-6 (r = 0.424, *p* < 0.001). These findings resonate with Bawankar et al., who reported elevated IL-1β and cortisol in smokers with periodontitis and noted that perceived stress levels were positively correlated with salivary cortisol [19]. In addition, Zhang et al. found statistically significantly higher levels of salivary cortisol and serum interleukin-1β in patients with periodontitis associated with smoking and stress [27].

From a psychoneuroimmunological perspective, chronic activation of the HPA axis and persistently elevated cortisol levels can have complex effects on immune function. While acute cortisol elevation generally suppresses inflammation, chronic exposure can result in glucocorticoid receptor desensitization or resistance, leading to impaired feedback inhibition and dysregulation of immune response [13,22,28]. This dysregulation may ultimately promote increased expression of pro-inflammatory cytokines such as IL-1β and IL-6, as has been demonstrated in experimental and clinical studies of chronic stress and inflammatory diseases [5,9,12,28]. Such mechanisms explain the observed associations between salivary cortisol and inflammatory markers in patients with severe periodontitis in our study.

While prior work by Cakmak et al. and Obulareddy et al. showed elevated cortisol in patients with aggressive and chronic periodontitis [6,31], our study contextualizes these elevations within a psychoneuroimmunological paradigm by concurrently assessing IL-1β and IL-6. An important element of novelty in our study is the detailed integration of cytokine and cortisol data with the 2017 Classification of periodontitis stages and grades. Only a few previous studies have examined cortisol across this classification spectrum [11,32,33].

In alignment with prior findings by Hingorjo et al. [11], we observed significantly higher cortisol levels in patients with advanced periodontal destruction, underscoring the role of stress-related endocrine response in modulating disease activity. Notably, our findings extend this relationship by demonstrating that salivary cortisol remains a robust independent predictor of both Stage III/IV and Grade C of periodontitis, even after adjusting for conventional confounders such as age, smoking, and inflammatory cytokines. While Dubar et al. highlighted cortisol’s association with periodontal pocket depth and microbial shifts, their study did not establish a predictive role for cortisol in disease classification [33]. Furthermore, unlike Petit et al., who evaluated plasma cortisol and its association with post-treatment outcomes [32], our study focused on salivary cortisol—a more practical, non-invasive biomarker—thus offering clinically applicable insights for risk stratification. The significant correlation between cortisol and IL-6 and IL-1β also suggests that specific inflammatory pathways may mediate the stress–periodontitis axis.

Our results also revealed significant associations between cortisol and core clinical periodontal parameters. Cortisol was positively correlated with BoP, PPD > 5 mm, and CAL ≥ 5 mm, and negatively with shallow pockets (PPD ≤ 3 mm) and mild attachment loss (CAL = 1–2 mm). These correlations suggest that cortisol reflects not merely disease presence but its clinical intensity.

Baumeister et al. reached similar conclusions in a large cohort of 3388 individuals, where serum cortisol was associated with greater mean CAL and increased BoP over time. However, their Mendelian randomization approach did not support a causal role of genetically predicted long-term cortisol exposure, implying that cortisol may be a marker of underlying systemic stress or inflammatory burden rather than a direct etiological agent [24]. This highlights that while such observational associations are valuable in clinical risk profiling and support the utility of cortisol in multifactorial assessment models, its elevation may arise as a consequence of chronic periodontal inflammation or systemic disease burden, rather than serve as a primary driver. Baumeister et al.’s use of genetic instruments to interrogate causality and the absence of supportive evidence therein specifically emphasize the importance of temporality in such associations [24]. Nevertheless, the consistent pattern of cortisol elevation alongside markers of disease severity and cytokine activity indicates a meaningful clinical relationship worthy of further longitudinal and interventional study.

Prior research also lends credibility to the predictive value of cortisol in treatment response. Villafuerte et al., in a systematic review, found that patients under chronic stress had a poorer response to non-surgical periodontal therapy (NSPT), with higher residual PPD and BoP post-treatment [34]. This study also noted that stressed patients had higher cortisol levels associated with periodontitis progression. Similarly, Petit et al. revealed that baseline stress predicted less favorable healing after scaling and root planing (SRP), particularly in deeper pockets [32]. However, the authors did not find a correlation between plasma cortisol and treatment outcomes. These findings, as well as the demonstrated elevated cortisol levels in Stage III/IV and Grade C periodontitis in our study, suggest that cortisol may serve as a valuable biomarker for identifying patients with an increased risk of stress-related disease progression. However, while psychological stress has been linked to poorer treatment outcomes, the direct prognostic role of cortisol for treatment response is not universally confirmed.

The relationship between cortisol levels, gender, and smoking in the context of periodontitis represents a complex area, with various studies revealing both consistent and contradictory results. Some research suggested that female gender may be a significant predictor for higher cortisol levels and cortisol/DHEA ratios, indicating a difference in HPA axis activity between sexes [8]. However, other studies have not found significant differences in cortisol levels between genders or have adjusted for gender as a potential confounding factor [20]. Regarding our study’s results, no statistically significant difference in cortisol levels was found based on gender (*p* = 0.259). Also, gender was not a significant predictor for severe periodontitis (OR = 1.55, *p* = 0.455). One possible explanation for our findings is the exclusion of participants using hormonal contraceptives or hormone replacement therapy, which may have minimized hormonal fluctuations and gender-related variability. Additionally, our sample size and cross-sectional design may also have limited the ability to detect more subtle gender differences.

As for smoking, it is widely recognized as a major risk factor for periodontitis [19,35,36]. Several studies supported an association between smoking and elevated cortisol levels in periodontitis patients [19,27]. For instance, smoking patients with chronic periodontitis demonstrated increased salivary cortisol concentrations when compared with non-smokers with the same condition [19,27]. Smoking can lead to HPA axis activation and affect the immune response, further exacerbating periodontal disease [3,19]. It has also been found that anxiety levels were higher in smokers, with a strong positive correlation between smoking duration and frequency and anxiety levels [36].

Despite these findings, other studies have not observed significant differences in cortisol concentrations between smokers and non-smokers [8,33]. Ansai et al. even suggested that smoking, as a stronger risk factor, might mask the relationship between cortisol and periodontitis severity in non-smokers [37]. In agreement with this, although our study found a significant difference in smoking status between Grade B and Grade C periodontitis patients, with 78.8% of Grade C patients being smokers versus 26.5% in Grade B (*p* < 0.001), smoking itself was not a statistically significant predictor for severe periodontitis (OR = 1.39, *p* = 0.333) in the adjusted multivariate model. These contradictions underscore the complexity of the dual mechanism—behavioral and physiological factors—through which stress and lifestyle factors, such as smoking, influence periodontal health and necessitate the need for further in-depth studies to clarify these interactions.

Despite the cross-sectional nature of this study, which precludes causal inferences, our findings that salivary cortisol is an independent predictor for severe periodontitis (Stage III/IV) and also for Grade C periodontitis may have some potential clinical implications. Although the odds ratios observed for salivary cortisol (OR = 1.23 and OR = 1.24, respectively) were statistically significant, their effect sizes are relatively modest. This suggests that, while cortisol may be valuable for clinical risk profiling and support its utility within multifactorial assessment models, it is likely to be only one component in the complex pathogenesis of periodontitis. Our results suggest that cortisol monitoring could contribute to identifying patients at increased risk for more severe forms or faster progression of periodontitis, independently of other demographic and behavioral risk factors.

To further evaluate the potential clinical applicability of salivary cortisol, we performed exploratory ROC curve analyses. These analyses revealed moderate diagnostic accuracy for both Stage III/IV vs. Stage I/II (AUC = 0.68, optimal threshold 11.57 nmol/L) and Grade C vs. Grade B periodontitis (AUC = 0.67, optimal threshold 9.76 nmol/L). While these thresholds provide insight into possible cut-off values for clinical use, the observed sensitivity (54% for Stage III/IV and 61% for Grade C) and specificity (88% for Stage III/IV and 76% for Grade C) indicate that salivary cortisol should be integrated into a broader risk assessment framework rather than used as a standalone diagnostic marker. The clinical utility of salivary cortisol as a predictive biomarker will require further validation in larger, more diverse cohorts, ideally as part of a broader panel of risk indicators.

Several limitations must be acknowledged. First, we did not use validated psychometric instruments such as the DASS-42 or Perceived Stress Scale (PSS), limiting our ability to correlate objective cortisol values with subjective stress levels. Second, we did not assess long-term periodontal outcomes, which restricts our interpretation of cortisol as a predictive biomarker. Third, information on socioeconomic status was not collected in this study, and its absence may represent an unmeasured confounding factor, as socioeconomic status can influence both psychological stress and periodontal health. Additionally, although we excluded women using hormonal contraceptives or hormone replacement therapy, we did not further record the specific phase of the menstrual cycle for female participants, which may have a potential influence on salivary cortisol levels. Future studies should address these gaps through longitudinal monitoring, inclusion of stress inventories, and interventional approaches such as stress-reduction protocols or behavioral therapies.

Some studies suggest that behavioral and psychological interventions—including motivational interviewing, cognitive behavioral therapy, and stress reduction practices such as yoga—may benefit patients with periodontitis by improving oral hygiene behaviors and reducing biological markers of stress [38,39,40]. However, while these findings are encouraging, the mechanisms by which stress reduction strategies impact periodontal health—whether through direct biological modulation (e.g., lowering cortisol and pro-inflammatory cytokines) or indirectly via improved health behaviors—remain an important area for further research.

## 5. Conclusions

This study provides a multidimensional analysis of the relationship between stress, inflammation, and periodontal disease, highlighting salivary cortisol as a promising biomarker reflective of both clinical severity and immunoinflammatory activity. By integrating cortisol with staging and grading of periodontitis and concurrently evaluating IL-1β and IL-6, we offer new insights into the complex biological underpinnings of stress-related periodontal pathology. Our findings further demonstrate that salivary cortisol has moderate discriminative ability for distinguishing between different stages and grades of periodontitis, with potential thresholds identified through ROC curve analysis. These results support the relevance of salivary cortisol as a non-invasive biomarker for periodontal disease risk stratification. Its clinical utility and optimal application should be further explored in future longitudinal and interventional studies, ideally within multifactorial predictive models and in broader patient populations.

## Figures and Tables

**Figure 1 medsci-13-00120-f001:**
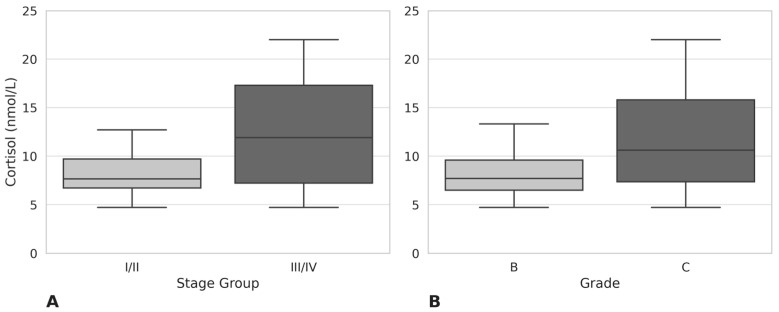
Salivary cortisol concentrations by periodontitis stage and grade. Box plots show salivary cortisol levels (nmol/L) in patients with (**A**) Stage I/II vs. Stage III/IV periodontitis; (**B**) Grade B vs. Grade C periodontitis. Group comparisons were assessed using the Mann-Whitney U test.

**Figure 2 medsci-13-00120-f002:**
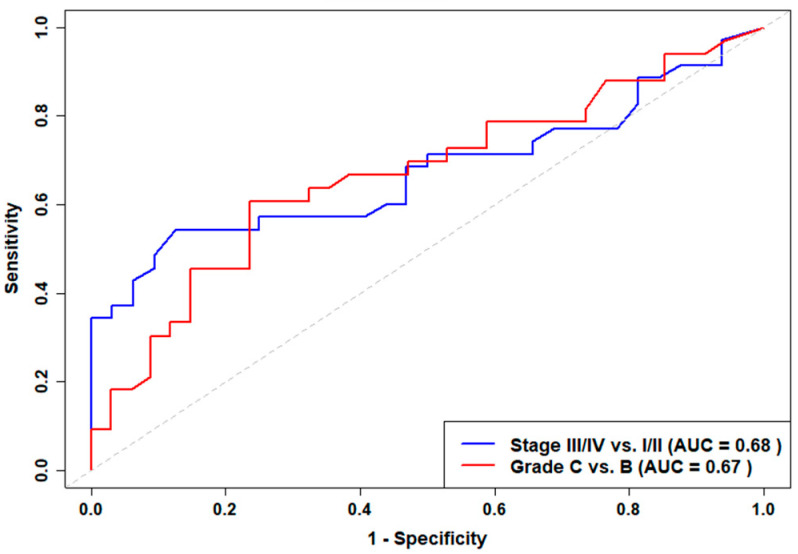
ROC curves for salivary cortisol distinguishing Stage III/IV vs. Stage I/II periodontitis (blue, AUC = 0.68) and Grade C vs. Grade B periodontitis (red, AUC = 0.67).

**Figure 3 medsci-13-00120-f003:**
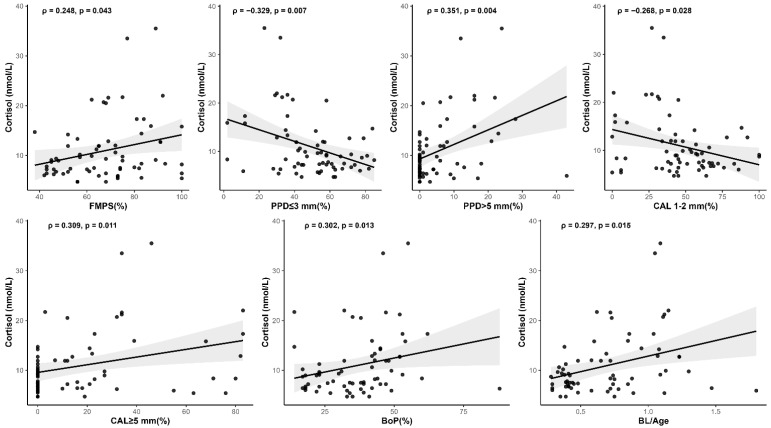
Scatter plots illustrate significant (*p* < 0.05) Spearman correlations between salivary cortisol and clinical periodontal parameters.

**Figure 4 medsci-13-00120-f004:**
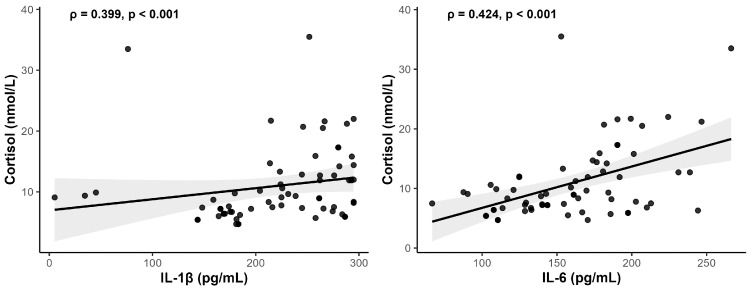
Scatter plots illustrate significant (*p* < 0.05) Spearman correlations between salivary cortisol and IL-1β and IL-6.

**Table 1 medsci-13-00120-t001:** Demographic characteristics by periodontitis stage and grade.

	Stage I/II (*n* = 32)	Stage III/IV (*n* = 35)	*p*-Value	Grade B (*n* = 34)	Grade C (*n* = 33)	*p*-Value
Mean Age ± SD	50.4 ± 10.5	52.3 ± 8.4	0.409	52.7 ± 10.1	50.1 ± 8.6	0.327
Females, *n* (%)	19 (59.4%)	16 (45.7%)	0.382	18 (52.9%)	17 (51.5%)	1.000
Males, *n* (%)	13 (40.6%)	19 (54.3%)		16 (47.1%)	16 (48.5%)	
Non-Smokers, *n* (%)	17 (53.1%)	15 (42.9%)	0.551	25 (73.5%)	7 (21.2%)	**<0.001**
Smokers, *n* (%)	15 (46.9%)	20 (57.1%)		9 (26.5%)	26 (78.8%)	

All data are displayed as mean ± standard deviation (SD) or as sample counts (*n*) and percentages (%). *p*-values are for comparisons between Stage I/II vs. III/IV and Grade B vs. Grade C, using a *t*-test for age and a chi-squared test for categorical variables. Statistically significant values (*p* < 0.05) are in bold.

**Table 2 medsci-13-00120-t002:** Multivariate logistic regression predicting severe periodontitis (Stage III/IV).

Predictor	Odds Ratio (OR)	95% CI	*p*-Value
Age	1.04	0.98–1.11	0.219
Gender	1.55	0.49–4.99	0.455
Smoking	1.39	0.72–2.81	0.333
Cortisol	1.23	1.07–1.45	**0.007**
IL-1β	1.01	1.00–1.02	0.210
IL-6	0.99	0.97–1.01	0.359

Odds ratios (OR) and 95% confidence intervals (CI). Statistically significant values (*p* < 0.05) are in bold.

**Table 3 medsci-13-00120-t003:** Multivariate logistic regression predicting periodontitis Grade C.

Predictor	Odds Ratio (OR)	95% CI	*p*-Value
Age	1.04	0.95–1.14	0.426
Gender	2.09	0.45–11.80	0.366
Smoking	13.6	4.7–61.7	**<0.001**
Cortisol	1.24	1.06–1.56	**0.026**
IL-1β	0.99	0.98–1.01	0.464
IL-6	1.01	0.99–1.04	0.337

Odds ratios (OR) and 95% confidence intervals (CI). Statistically significant values (*p* < 0.05) are in bold.

**Table 4 medsci-13-00120-t004:** Significant associations between salivary biomarker concentrations and periodontal clinical parameters.

Biomarker	Clinical Parameter	Spearman’s ρ	*p*-Value
**Cortisol**	FMPS (%)	0.248	0.043
	PPD ≤ 3 mm (%)	−0.329	0.007
	PPD > 5 mm (%)	0.351	0.004
	CAL 1–2 mm (%)	−0.268	0.028
	CAL ≥ 5 mm (%)	0.309	0.011
	BoP (%)	0.302	0.013
	BL/Age	0.297	0.015
**IL-1β**	FMPS (%)	0.320	0.008
	FMBS (%)	0.296	0.015
	PPD ≤ 3 mm (%)	−0.338	0.005
	PPD > 5 mm (%)	0.278	0.023
	CAL 1–2 mm (%)	−0.340	0.005
	CAL ≥ 5 mm (%)	0.250	0.041
	BoP (%)	0.469	<0.001
	BL/Age	0.312	0.010
**IL-6**	PPD ≤ 3 mm (%)	−0.258	0.035
	BoP (%)	0.262	0.032
	BL/Age	0.265	0.030

Displayed are correlations reaching statistical significance (*p* < 0.05), as determined by Spearman’s rank correlation coefficient (ρ).

## Data Availability

The data presented in this study are available upon request from the corresponding author due to privacy and ethical requirements.

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
