# Peer review of "Salivary Cortisol and Periodontitis Severity: A Cross-Sectional Biomarker-Based Assessment of Stress and Inflammation"

_medsci, 2025, doi:10.3390/medsci13030120_

Round 1
Reviewer 1 Report
Comments and Suggestions for Authors Although the relationship between psychological stress, inflammatory cytokines, and periodontitis have been investigated, there is few study evaluating standardized periodontal disease staging and grading with salivary cortisol, interleukin-1beta, and interleukin-6. I think the topic of this study is original. The conclusions are consistent with the evidence and arguments presented and answer the key question. There are no additional comments on figures and tables. I think it is suitable for publication to this journal in the original form.Author Response
We sincerely thank you for your positive evaluation of our manuscript and for recognizing the originality of our research topic, as well as the consistency and clarity of our conclusions. We appreciate your supportive comments regarding the study’s design, findings, and presentation. Thank you for recommending our work for publication!
Reviewer 2 Report
Comments and Suggestions for Authors
The authors conducted a cross-sectional observational study to evaluate salivary levels of cortisol, interleukin-1β (IL-1β), and interleukin-6 (IL-6) in relation to periodontitis severity. Sixty-seven patients were categorized by Stage (I/II vs III/IV) and Grade (B vs C). Biomarkers were measured using ELISA and electrochemiluminescence assays. Results showed significantly higher cortisol levels in more advanced disease stages and grades. Cortisol independently predicted severe periodontitis and higher progression risk, while inflammatory markers were associated with clinical periodontal parameters. The study concludes that salivary cortisol may be a useful non-invasive biomarker for risk stratification in periodontitis.
Although the manuscript is interesting several weaknesses were identified.
Specific comments are noted below:
The title omits the type of study.
The abstract lacks specific numerical data (e.g., cortisol levels, p-values) that support the significance of findings.
Moreover, the word “promising” in “salivary cortisol is a promising biomarker…” introduces an element of speculation.
The introduction overrelies on theoretical backgroundregarding cortisol’s effects without balancing it with critiques of inconsistent evidence in the literature.
Besides, the authors do not clearly justify why IL-6 and IL-1β were selected over other established biomarkers.
Also, the hypothesis but would benefit from explicitly stating the primary and secondary objectiveswith more precision.
Materials and Methods
Sample size justification is absent.
Furthermore, potential confounders, such as socioeconomic status, oral hygiene habits, and systemic inflammation, are not included in the analysis or discussion.
The timing of saliva collection does not mention whether collection was standardized across the menstrual cycle in female patients—cortisol levels can vary significantly.
The multivariate models include variables without clarity on the method of selection (e.g., stepwise vs. enter method).
Results
The results related to IL-1β and IL-6 are underreported.
Also, the use of scatter plots are referenced without figure numbers, reducing clarity for readers.
Figure legends should clarify what statistical comparisons were made.
Table 4 could include non-significant correlations for completeness or justify their exclusion.
Some effect sizes (e.g., odds ratios for cortisol) are modest but presented with emphasis without discussion of their clinical relevance.
The discussion at times overstates causality.
Furthermore, while limitations are mentioned, the absence of psychometric stress instruments is downplayed, despite being a major gap when discussing "psychological stress."
Also, the role of gender is addressed, but the inconsistencies in the literature are cited without critically analyzing why the present study found no effect.
Smoking is identified as an independent predictor for Grade C, but not for Stage III/IV, yet this discrepancy is not explained.
The reference to psychoneuroimmunology lacks depth in connecting how chronic cortisol elevation may biologically drive IL-1β or IL-6 expression.
In the conclusions, the statement “salivary cortisol is a promising non-invasive biomarker” is too strong given the observational nature of the study and the limited external validation.
Moreover, the conclusions mention risk stratification potential, but this is speculative as no predictive model or ROC analysis was conducted.
Some cited studies are not international in scope, which could limit generalizability of contextual claims.
Author Response
Thank you for your time and valuable feedback on our manuscript. Your suggestions have helped us further improve the clarity and completeness of our study. Please find our detailed responses below. All revisions have been incorporated into the revised manuscript and highlighted in yellow.
Point-by-point response to Comments and Suggestions for Authors:
Comments 1: The title omits the type of study.
Response 1: We agree that including the type of study will improve the clarity of the manuscript. Accordingly, we have revised the title to:
“Salivary Cortisol and Periodontitis Severity: A Cross-Sectional Biomarker-Based Assessment of Stress and Inflammation”.
Comments 2: The abstract lacks specific numerical data (e.g., cortisol levels, p-values) that support the significance of findings.
Response 2: Thank you for this suggestion. In response, we have revised the abstract to include key numerical results, such as median cortisol levels and relevant p-values, to better support the significance of our findings (on page 1, lines 22-27).
Comments 3: Moreover, the word “promising” in “salivary cortisol is a promising biomarker…” introduces an element of speculation.
Response 3: Thank you for this observation. We acknowledge the importance of avoiding speculative language. However, we chose to use the word “promising” to convey the potential of salivary cortisol as a biomarker, while making it clear in the following sentence that further validation in longitudinal studies is required. This structure was intended to communicate both the potential value of our findings and the need for future research before clinical implementation. We believe this approach aligns with common scientific practice, where the term “promising” is used to highlight potential while explicitly recognizing current limitations.
Comments 4: The introduction overrelies on theoretical background regarding cortisol’s effects without balancing it with critiques of inconsistent evidence in the literature.
Response 4: Thank you for your comment. We agree that it is important to acknowledge inconsistent findings in literature. In the Introduction section (on page 2, lines 71-78), we have highlighted the conflicting results from previous studies, noting that while some report elevated salivary cortisol in periodontitis, others have found no significant differences. We also discuss methodological factors contributing to this heterogeneity, such as study design and diagnostic criteria.
Comments 5: Besides, the authors do not clearly justify why IL-6 and IL-1β were selected over other established biomarkers.
Response 5: Thank you for this observation. In our study, we selected IL-1β and IL-6 because these cytokines are widely recognized in periodontal research as central mediators of inflammation and tissue destruction. As described in the Introduction, our aim was to provide a comprehensive assessment of both neuroendocrine (cortisol) and inflammatory (IL-1β, IL-6) markers to better understand the interplay between stress and immune response in periodontitis. We have added the following sentence (on page 3, lines 96-98): “In this study, we focused on IL-1β and IL-6, which are central inflammatory mediators in periodontitis, to provide a representative assessment of the inflammatory response alongside salivary cortisol”.
Comments 6: Also, the hypothesis would benefit from explicitly stating the primary and secondary objectives with more precision.
Response 6: To address your comment, we have revised the hypothesis section to state our research hypothesis and objectives more clearly and precisely within the narrative flow (on page 3, lines 102-106).
Comments 7: Sample size justification is absent.
Response 7: Thank you for this important comment. To address your concern, we performed a post hoc power analysis for the primary comparisons of salivary cortisol levels between Stage I/II and Stage III/IV, as well as between Grade B and Grade C periodontitis groups. We have added this information in the Materials and Methods section, Statistical Analysis subsection (on page 4, lines 164-170).
Comments 8: Furthermore, potential confounders, such as socioeconomic status, oral hygiene habits, and systemic inflammation, are not included in the analysis or discussion.
Response 8: Thank you for this thoughtful comment. We would like to clarify that oral hygiene was objectively assessed in our study using the Full Mouth Plaque Score (FMPS), which was analyzed both as a clinical parameter and in relation to salivary biomarkers. Regarding systemic inflammation, while we did not assess classic systemic markers such as CRP, we measured IL-1β and IL-6 in saliva, which are established inflammatory biomarkers relevant to periodontal disease. However, we acknowledge that socioeconomic status was not collected and thus could not be included in our analysis. We have now expanded the limitations section to discuss the absence of socioeconomic data (on page 12, lines 442-444).
Comments 9: The timing of saliva collection does not mention whether collection was standardized across the menstrual cycle in female patients—cortisol levels can vary significantly.
Response 9: Thank you for raising this important point. We acknowledge the potential influence of the menstrual cycle on salivary cortisol levels. In our study, we excluded women using hormonal contraceptives or hormone replacement therapy, as stated in the exclusion criteria, to minimize external hormonal influences. However, we did not further record the specific phase of the menstrual cycle for female participants not using these medications. We have acknowledged this as a limitation in the Discussion (on page 12, lines 445-447).
Comments 10: The multivariate models include variables without clarity on the method of selection (e.g., stepwise vs. enter method).
Response 10: Thank you for pointing out this omission. In our analysis, all variables were entered simultaneously into the multivariate logistic regression models, based on their known or potential relevance to periodontitis severity and progression. We have clarified this approach in the revised Methods section (on page 4, lines 161-163).
Comments 11: The results related to IL-1β and IL-6 are underreported.
Response 11: We respectfully note that while our primary outcome was salivary cortisol, we have provided descriptive statistics (medians, IQRs, and p-values) and relevant statistical associations for IL-1β and IL-6 by stage and grade, as well as their correlations with clinical periodontal parameters in both the results text (on page 5, lines 202-208) and Table 4 on pages 6 and 7. In line with our study objectives, we aimed to report these findings clearly and concisely.
Comments 12: Also, the use of scatter plots are referenced without figure numbers, reducing clarity for readers.
Response 12: Thank you for your comment. We have reviewed the manuscript and confirm that scatter plots are referenced with figure numbers (e.g., Figure 2 and Figure 3) in the Results section to ensure clarity for readers.
Comments 13: Figure legends should clarify what statistical comparisons were made.
Response 13: We have revised the figure legends to clearly specify the statistical comparisons performed for each figure.
Comments 14: Table 4 could include non-significant correlations for completeness or justify their exclusion.
Response 14: Thank you for your comment. We intentionally reported only statistically significant correlations in Table 4 to enhance clarity and focus for readers (on page 7, lines 250-253). We believe this approach highlights the most relevant findings.
Comments 15: Some effect sizes (e.g., odds ratios for cortisol) are modest but presented with emphasis without discussion of their clinical relevance.
Response 15: Thank you for this important observation. We agree that while the odds ratios for cortisol were statistically significant, their magnitude is relatively modest and should be interpreted with caution. In response, we have modified the Discussion to acknowledge the modest effect sizes and to address their potential clinical relevance and limitations (on page 11, lines 428-432). We note that even small effect sizes can be meaningful in multifactorial diseases such as periodontitis, but that the predictive value of salivary cortisol as a biomarker should be considered within the broader context of other risk factors and patient characteristics.
Comments 16: The discussion at times overstates causality.
Response 16: We agree that, given the cross-sectional design of our study, causal inferences should be avoided. We have carefully revised statements that could be interpreted as causal, rephrasing them to emphasize association rather than causation. We have also reiterated the study’s cross-sectional nature and the need for longitudinal research to establish causality.
Comments 17: Furthermore, while limitations are mentioned, the absence of psychometric stress instruments is downplayed, despite being a major gap when discussing "psychological stress."
Response 17: We agree that the absence of validated psychometric instruments is a limitation, particularly in studies of psychological stress. However, we intentionally focused on objective biological markers (salivary cortisol and inflammatory cytokines) rather than self-reported instruments, as the latter can be prone to reporting bias and subjectivity. While this approach provides valuable, unbiased physiological data, we acknowledge that the lack of subjective stress assessment restricts our ability to directly correlate biological findings with perceived psychological stress.
Comments 18: Also, the role of gender is addressed, but the inconsistencies in the literature are cited without critically analyzing why the present study found no effect.
Response 18: Thank you for this insightful comment. We agree that further analysis is needed regarding the lack of gender differences in our study, especially in light of the inconsistencies reported in the literature. In the revised Discussion, we now provide a more critical perspective, suggesting possible reasons for our findings. Specifically, our study’s exclusion of participants using hormonal contraceptives or hormone replacement therapy, as well as the standardization of saliva collection timing, may have reduced hormonal variability and minimized gender effects on cortisol levels. Additionally, our sample size and the cross-sectional design may limit the detection of subtle gender-related differences. We have added these considerations to the Discussion section for greater clarity (on page 11, lines 401-405).
Comments 19: Smoking is identified as an independent predictor for Grade C, but not for Stage III/IV, yet this discrepancy is not explained.
Response 19: Thank you for this valuable comment. In our study, smoking emerged as an independent predictor for Grade C periodontitis, but not for Stage III/IV disease. This difference likely reflects the current classification system, where “grade” incorporates risk factors such as smoking and diabetes, while “stage” is determined primarily by the extent and severity of periodontal tissue destruction at a single point in time. Therefore, smoking status has a direct influence on the assignment of Grade C but may have less impact on staging, especially after adjusting for other variables in multivariate analysis. We have clarified this point in the Results section (on page 6, lines 224-227).
Comments 20: The reference to psychoneuroimmunology lacks depth in connecting how chronic cortisol elevation may biologically drive IL-1β or IL-6 expression.
Response 20: We agree that the connection between chronic cortisol elevation and the expression of inflammatory cytokines such as IL-1β and IL-6 could be further elaborated. In the revised Discussion, we have expanded on the psychoneuroimmunological mechanisms that may underlie these associations (on page 9, lines 330-339).
Comments 21: In the conclusions, the statement “salivary cortisol is a promising non-invasive biomarker” is too strong given the observational nature of the study and the limited external validation.
Response 21: We acknowledge the importance of avoiding speculative language. However, we intentionally used the word “promising” to convey the potential of salivary cortisol as a biomarker, and we made it clear that further investigation in longitudinal studies is required. This structure was meant to reflect both the potential value of our findings and the necessity for additional research before clinical application. We believe this wording is consistent with standard scientific practice, where the term “promising” is often used to indicate potential utility while explicitly recognizing current limitations.
Comments 22: Moreover, the conclusions mention risk stratification potential, but this is speculative as no predictive model or ROC analysis was conducted.
Response 22: We agree that statements regarding the potential of salivary cortisol for risk stratification are speculative in the absence of predictive modeling or ROC analysis. We have revised the conclusions to avoid overstatement and have clarified that further research – including predictive modeling and ROC analysis – is needed to evaluate the risk stratification value of salivary cortisol in periodontitis (on page 12, lines 466-469).
Comments 23: Some cited studies are not international in scope, which could limit generalizability of contextual claims.
Response 23: We appreciate the reviewer’s observation. While our references include both regional and international studies, we have aimed to interpret findings in light of their respective contexts and limitations.

Reviewer 3 Report
Comments and Suggestions for Authors
This study aimed to compare salivary levels of cortisol, interleukin-1β, and interleukin-6 among patients with different severities of periodontitis and to assess their associations with clinical periodontal parameters. A total of 67 patients were enrolled, providing a reasonable sample size. The findings showed that salivary cortisol independently predicted both Stage III/IV periodontitis (OR = 1.23, p = 0.007) and Grade C periodontitis (OR = 1.24, p = 0.026). The study is clinically relevant. As stress is difficult to quantify but has significant influence over periodontal health. The manuscript is of interest, but some comments as follows:
1. Since the main topic is also about stress, why the stress levels of the patients were not measured in the first place? And as such limitation was discussed, some statements about "stress" should be toned down.
2. Could the author showed a representative case with high level of cortisol? Even the radiographes will be helpful in engaging readers.
3. Please also discuss potential approaches in addressing patients' stress
4. Since smoking is defining the grading, why is the author calculating its predictive value?
5. Please also discuss about the threshold for the cotisol level for future translation.
Author Response
We sincerely thank you for your thoughtful and constructive feedback on our manuscript. We appreciate the recognition of our work and have addressed each of the suggestions carefully to improve the clarity of our manuscript. Our detailed responses are provided below. All changes have been incorporated into the revised manuscript and highlighted in yellow.
Point-by-point response to Comments and Suggestions for Authors:
Comments 1: Since the main topic is also about stress, why the stress levels of the patients were not measured in the first place? And as such limitation was discussed, some statements about "stress" should be toned down.
Response 1: Thank you for this important comment. We acknowledge that, while stress is a central focus of our study, we did not include a validated psychometric instrument to measure subjective stress levels among participants. Our study design prioritized the use of objective biological markers (salivary cortisol and inflammatory cytokines), but we recognize that this approach does not capture the subjective experience of psychological stress. As already discussed in the limitations section, the absence of a psychometric stress assessment restricts our ability to directly correlate biological findings with perceived stress (on page 12, lines 438-440).
Comments 2: Could the author show a representative case with high level of cortisol? Even the radiographes will be helpful in engaging readers.
Response 2: Thank you for your suggestion. While we agree that presenting a representative clinical case can be valuable for illustrating disease severity, our study was designed as a cross-sectional, biomarker-based analysis focused on group-level associations rather than individual case reporting. We believe that the inclusion of a single case may not be fully representative of the broader findings or in line with the primary objectives and methodology of our study. However, we appreciate the idea and agree that future work – such as a case series or longitudinal follow-up of patients with high salivary cortisol levels – would be appropriate to illustrate clinical-radiographic and biomarker dynamics in more detail.
Comments 3: Please also discuss potential approaches in addressing patients' stress.
Response 3: Thank you for this valuable suggestion. We have now integrated a concise discussion of recent evidence on stress management approaches in periodontitis into the manuscript (on page 12, lines 451-458). Given our cross-sectional design, we did not assess specific interventions, but we now highlight this as a promising direction for future longitudinal and interventional studies.
Comments 4: Since smoking is defining the grading, why is the author calculating its predictive value?
Response 4: We agree that smoking is an integral factor in the current periodontitis grading system, as outlined in the 2017 World Workshop classification. Our intention was to evaluate the relative contribution of known risk factors – including smoking – within our cohort for both stage and grade of periodontitis. Therefore, its significant association with Grade C in our analysis is expected and reflects the structure of the grading system. We have clarified this in the Results section and have avoided interpreting this as a novel finding (on page 6, lines 224-227).
Comments 5: Please also discuss about the threshold for the cortisol level for future translation.
Response 5: Thank you for this suggestion. Although our results demonstrated significant group differences in salivary cortisol levels, our study design and sample size do not allow us to define a clinical threshold. We have clarified this point in the revised Conclusions section (on page 12, lines 466-469) and highlight the need for future research, including ROC analysis, to establish validated cut-off values for clinical use.

Round 2
Reviewer 2 Report
Comments and Suggestions for Authors
No further comments
Author Response
Thank you for your positive assessment of our manuscript. We appreciate your time and constructive evaluation.
Reviewer 3 Report
Comments and Suggestions for Authors
Thank you for addressing most of the comments and revising the manuscript. It is a solid study. Yet adding clinical records and "discuss" about the range for the cortisol level will appeal to the clinicians and project for future translation.
Author Response
Thank you for this valuable suggestion and your positive feedback on our study. In response, we have added inter-group ROC analyses within our periodontitis cohort, as well as a discussion of optimal cortisol cut-off values, to the Methods (on page 4, lines 166-171), Results (on page 7, lines 252-262), and Discussion (on page 12, lines 454-461) sections, highlighting their potential clinical relevance and translational significance. The added paragraphs are highlighted in turquoise color in the revised manuscript for your convenience.
Round 3
Reviewer 3 Report
Comments and Suggestions for Authors
Thank you for addressing the comments